# Decreased Need for Correction Boluses with Universal Utilisation of Dual-Wave Boluses in Children with Type 1 Diabetes

**DOI:** 10.3390/jcm11061689

**Published:** 2022-03-18

**Authors:** Mari Lukka, Vallo Tillmann, Aleksandr Peet

**Affiliations:** 1Department of Paediatrics, Institute of Clinical Medicine, University of Tartu, 50406 Tartu, Estonia; vallo.tillmann@kliinikum.ee (V.T.); aleksandr.peet@kliinikum.ee (A.P.); 2Children’s Clinic of Tartu University Hospital, 51014 Tartu, Estonia

**Keywords:** type 1 diabetes, insulin pump therapy, CGM, meal bolus, dual wave bolus, standard bolus

## Abstract

Insulin pumps offer standard (SB), square and dual-wave boluses (DWB). Few recommendations exist on how to use these dosing options. Several studies suggest that the DWB is more effective for high-fat or high-carbohydrate meals. Our objective was to test whether time in range (TIR) improves in children with type 1 diabetes (T1D) using the universal utilization of the dual-wave boluses for all evening meals regardless of the composition of the meal. This was a 28-day long prospective randomized open-label single-center crossover study. Twenty-eight children with T1DM using a Medtronic 640G pump and continuous glucose monitoring system were randomly assigned to receive either DWB or SB for all meals starting from 6:00 p.m. based solely on the food carbohydrate count. DWB was set for 50/50% with the second part extended over 2 h. After two weeks patients switched into the alternative treatment arm. TIR (3.9–10 mmol/L), time below range (TBR) (<3.9 mmol/L) and time above range (TAR) (>10 mmol/L) and sensor glucose values were measured and compared between the groups. Twenty-four children aged 7–14 years completed the study according to the study protocol. There were no statistically significant differences in mean TIR (60.9% vs. 58.8%; *p* = 0.3), TBR (1.6% vs. 1.7%; *p* = 0.7) or TAR (37.5 vs. 39%; *p* = 0.5) between DWB and SB groups, respectively. Subjects in the SB treatment arm administered significantly less correction boluses between 6 p.m. and 6 a.m. compared to those in the DWB group (1.2 ± 0.8 vs. 1.7 ± 0.8, respectively; *p* < 0.01). DWB for evening meals in which insulin is calculated solely on the food carbohydrate content did not improve TIR compared to standard bolus in children with T1D. However, DWB enabled to use significantly less correction boluses to achieve euglycemia by the morning compared to the SB.

## 1. Introduction

Despite of the wide use of modern diabetes technology among the pediatric population in the last decade [1] metabolic control still remains suboptimal in many cases [2]. One of the main obstacles preventing the achievement of treatment goals is controlling postprandial hyperglycemia. Information about how to manage glycemic excursions after meals is relatively limited. It is well known that the timing of the meal boluses is essential [3], but when high-fat or high-carbohydrate meals are consumed, optimal postprandial glycaemia is still hard to achieve [4].

Modern insulin pumps offer a variation of preprogrammed boluses: standard bolus (SB) square and dual-wave boluses (DWB). However, only few recommendations exist on how to benefit from the use of different bolus administration types [5]. The International Society for Pediatric and Adolescent Diabetes states that the impact of dietary fat and protein should be considered when determining the insulin bolus dose and delivery [5], but the optimal insulin bolus dose for meals high in fat and protein is undefined [4].

The use of a fat-protein unit has been previously proposed [6], but the method seems too complex for everyday use for most patients and was associated with a higher rate of hypoglycemia [7]. According to previous research the prolonged bolus administration methods might have certain advantages for high-fat or high-carbohydrate meals [3,8,9,10,11]. For example, Jones et al. showed that the 8-h DWB was superior compared to the single wave bolus and provided the best glycemic control after a pizza meal [9]. Chase et al. demonstrated that the dual wave option where 70% is administered as a standard bolus and 30% as a square-wave over 2 h provided the most effective method of insulin administration for a meal high in carbohydrates, fat and protein [10]. O’Connell et al. confirmed the superiority of the dual-wave bolus (50% and 50% over 2 h) for low glycemic index foods as well [11]. These data suggest the DWB utilization might be the preferred insulin delivery mode as it resembles the physiologic postprandial biphasic insulin secretion described first by Curry et al. [12]. Optimal distribution of the DWB parts and duration of the extended part seems to be dependent on specific food content. Lopez et al. proposed that meals with a high fat and protein content require at least 60% of the total insulin upfront to control the initial postprandial glucose rise [13]. To our best knowledge there have been no controlled randomized studies investigating the benefit of one bolus type over the other in a real-life setting over an extended period. Previous studies have been mostly conducted in hospital-based settings and limited to analyze the effect of specific foods on postprandial hyperglycemia in very few individuals [8,9,10].

Therefore, we decided to perform a randomized controlled crossover study comparing SB to the DWB in a real-world setting. Considering that carb counting is difficult to master for many of our patients and substantial inter-individual differences exist in insulin dose requirements for fat and protein [14,15,16], high glycemic index and low glycemic index carbs [4], we decided to calculate the insulin dose solely based on the carbohydrate content in the food. Therefore, we decided to use DWB with a 50%/50% proportion with the second part extended for 2 h in order to achieve optimal insulin coverage for a typical Western type meal consumed for dinner. The main objective of the study was to test the hypothesis that the universal utilization of the DWB for all meals starting from 6:00 p.m. for 2 weeks improves time in range (TIR) in children with type 1 diabetes (T1D) compared to the SB use. The decision to limit the intervention period only to evening meals was carried out from the practical reason as this is the time when parents could supervise their children and follow the study plan.

## 2. Materials and Methods

This was a 28-day long prospective randomized open-label single-center crossover clinical study with a 14-day long run-in phase. Recruitment was carried out during an outpatient visit. All subjects and parents signed an informed consent form approved by the Research Ethics Committee of The University of Tartu before entering the study. The trial was registered in ClinicalTrial.gov under the identifier: NCT04668612. We used the following inclusion criteria: age 7–18 years, T1D duration over a one year, CGM and insulin pump treatment for at least 3 months prior to the recruitment, estimated HbA1c based on the 14-days CGM report below 8.5%, and a daily insulin dose of more than 0.5 international units per kilogram. A power-analysis was conducted and according to Rigby et al. the minimum sample size to demonstrate a clinically relevant change in TIR (10%) would have been 16 subjects [17]. All patients used the Medtronic MiniMed 640G pump with the Enlite sensors, because this system was most commonly used in our diabetic center at that time and equipped with the basal insulin auto suspend before low function. This was turned on in all study participants at the same level: insulin administration was temporarily suspended when the blood sugar of 3.9 mmol/L was predicted. Subjects with known diabetes complications or with elevated tissue transglutaminase IgA antibodies in the last two years and children who developed acute viral infections during the week preceding the recruitment were excluded. During the first study visit, which was the only on-site visit, we collected clinical data of subjects (age, gender, height, body weight, body mass index, pubertal stage according to Tanner scale) and evaluated the eligibility for the study. We titrated insulin doses, set up a bolus wizard in their pump and instructed patients accordingly. After the first study visit patients entered the run-in phase for two weeks. During the run-in period we optimized treatment as much as possible and titrated insulin doses twice, based on the patients CGM reports of the preceding week sent to us via e-mail. Thereafter patients were re-evaluated for eligibility according to the study inclusion and exclusion criteria. We implemented two additional exclusion criteria: patients with a basal insulin proportion of more than 55% or who developed an acute viral infection during the run-in phase, were excluded. 

After that, patients were randomly assigned into DWB or SB arms. After 14 days subjects switched into the alternative treatment arm. The study structure is shown in Figure 1. In the subsequent 2 weeks subjects received either DWB (50% of the insulin delivered as a bolus 10–15 min prior to the meal and 50% delivered over two hours) or a SB (100% of the insulin delivered as a bolus 10–15 min prior to the meal) for every meal after 06:00 p.m. Repeated meals were allowed provided mealtimes took place at least two hours apart. Eating with a shorter interval was only allowed in case of hypoglycemia <3.9 mmol/L. In case of hyperglycemia above 12 mmol/L for longer than 2 h after the start of the food bolus administration a correction bolus via standard bolus for both arms was recommended to the target value of 7.0 mmol/L. Prior to the study participants were not provided with specific instructions about CGM arrow trend management. Participants were encouraged to stick to their regular meal schedule and typical diet during the different treatment periods. During the 28-day long intervention period patients and parents were discouraged to change their basal doses on their own. 

The use of temporary basal was not recommended during 06:00 p.m. to 06:00 a.m. Within the intervention period correction bolus doses and food bolus coefficients were not modified, basal rate was adjusted only in the case of definite necessity as judged by the patient’s physician. After receiving two weeks of SB or DWB CGM data from the pump were downloaded at home and sent to the investigators. Time in range (TIR), (>3.9 mmol/L and below 10 mmol/L), time below range (TBR) (<3.9 mmol/L) and time above range (TAR) (>10 mmol/L) range were recorded from the pump after finishing 2 weeks of each bolus type. Compliance to the study terms was evaluated before the data analysis. Data of the study subject was only included to our analysis if at least 80% of boluses were administered according to the bolus type the patient was randomized to CGM data were analyzed with the Care Link Professional software, which is provided by the manufacturer for the Medtronic system users. Comparison of TIR, TBR and TAR between the two groups was performed using Student’s *t*–test. *p*-value 0.05 or less was considered as significant. The mean area under the curve (AUC) for the period from the first evening meal bolus after 6:00 p.m until 6:00 a.m. was calculated by the trapezoidal method [18]. The Statistical analysis was performed with R version 4.1.0 for Windows.

## 3. Results 

Out of the screened 32 patients, 31 were enrolled and gave written consent, 1 did not meet the inclusion criteria. In total 31 participants entered the run-in-period, 3 patients had problems uploading sensor data at home and thus 28 patients completed the run-in and underwent randomization. Four participants cancelled the study’s active arm due to the failure of following the study plan. Twenty-four patients (10 boys) with T1D, 7–14 years completed the study and their data was included into the analysis. The average age of the study subjects was 10.8 ± 2.0 years (Table 1). Before the run-in period, estimated HbA1c provided by the sensor CGM report was 7.6 ± 0.7%. Mean basal insulin proportion was 37.8 ± 10.7% at study entry. The average daily insulin dose per kilogram of the participants was 0.8 ± 0.1 IU/kg/d. After run-in and before randomization the estimated HbA1c of the study participants reduced to the 7.4 ± 0.5% (57.4–5.2 mmol/mol) and average basal insulin proportion remained almost unchanged on the level of 39 ± 10.5%. Mean TIR, mean TBR and mean TAR was not significantly different between the treatment arms as shown in Figure 2. The mean AUC calculated for the period from the first evening meal bolus after 6:00 p.m. until 6:00 a.m. was not statistically different between the SB and DWB study groups (98.73 ± 12.4 mmol/L × h vs. 101,63 ± 14.8 mmol/L × h, respectively; *p* = 0.243). Mean estimated HbA1c after two weeks of DWB or SB usage did not differ either and was 7.4 ± 0.5% for both treatment arms.

Patients in the DWB treatment arm administered significantly less correction boluses between 6 p.m. and 6 a.m. than in the SB arm (1.2 ± 0.8 vs. 1.7 ± 0.8; *p* < 0.01). Eight patients in the SB had to use at least 2 correction boluses during the night, whereas in the DWB group only four. The mean glucose profiles in both treatment arms are shown in Figure 3. The only statistically significant difference in mean glucose levels was seen 1 h after administering the bolus when it was significantly higher in the DWB arm compared to the SB arm (9.8 ± 1.6 mmol/L vs. 8.9 ± 1.8 mmol/L; *p* = 0.01), but with no differences at 2–6 h after the bolus. The mean number of basal suspension episodes from 6:00 PM to 06:00 AM did not differ between the DWB and SB treatment arms (0.99 ± 0.5 vs. 1 ± 0.6, respectively *p* = 0.89), neither were different the mean cumulative duration of suspension (2 h 27 min vs. 2 h 28 min, respectively; *p* = 0.887). Four patients in the DWB treatment arm improved their TIR be 14% to 23% whereas two patients increased their TIR by 13% and 14% in the SB treatment arm.

## 4. Discussion 

We described the effect of the systematic use of the DWB on the TIR, TBR and TAR parameters in pediatric T1D patients in a real-life setting. This study was not able to demonstrate the direct benefit of such universal usage of the DWB instead of the SB for the evening meals. These results might indicate the non-superiority of one bolus type compared with the other, but could also derive from the small effect of the evening boluses on the TIR, which is affected by other glycemic and bolusing events during the day as well. In order to identify the preferred bolus type we also calculated the mean AUC for postprandial blood sugar excursions, but this was not different between the treatment arms. However, it should be taken into an account that in the current study design where additional meals and correction boluses were allowed and used 2 h after the main evening meal the mean AUC is also not an ideal parameter to compare treatment arms. In this study we showed a significantly reduced need for correction boluses in the DWB treatment arm. 

Many authors have proposed that postprandial hyperglycemia is controlled more effectively when the insulin dose is calculated for both the carbohydrates and fat-protein nutrients [4,5,6,7] yet calculating the fat protein unit has not become standard practice since today although suggested a while ago. The main cause for this might be the complexity of this calculation and unwillingness of the patients to use this formulation for their prandial insulin dose. Until today the specific instructions are lacking for the accurate fat-protein adjustments and therefore we did not cover fat and proteins in our study. We hypothesized that the DWB, when calculated solely on the carbohydrate content of food, might be more effective than a SB in controlling late postprandial hyperglycemia, because it also covers the late effects of fats and proteins on blood sugar [4]. Several previously published studies support our suggestion, showing that postprandial hyperglycemia is more effectively controlled, when prolonged methods of insulin administration are used instead of the SB [8,9,10,13]. However, these conclusions are largely based on blood glucose analyses in very few individuals after single meal analysis in relatively controlled settings and the long-term benefit of the prolonged bolus types’ utilization has not been demonstrated. Bell et al. proposed that to achieve target glucose control following a high fat and high protein meal in adults a 30%/70% split over 2.4 h is required [14]. Lopez et al., when studying different split variations after a meal high in carbohydrates, calories and fat, demonstrated that a SB controlled the blood glucose excursion for the first 120 min only, after which there was a progressive rise in glucose level [13]. We also found that the SB provides significantly better control over the first hour of postprandial hyperglycemia, but in contrast to some previous studies [6,12], no difference in sensor glucose level was evident between the treatment arms at 2–6 h after the main evening meal bolus. Bell et al. showed that the optimal combination bolus split to maintain postprandial glycemia for a high-fat and high-protein meal was 60/40% or 70/30% delivered over 3 h [14], therefore our 50/50% split choice over 2 h might explain these differences in the results. Still in our study late hyperglycemia might have been controlled more effectively in the DWB arm, because they required a significantly lower number of correction boluses compared to the SB treatment arm to achieve normoglycemia by the morning. The difference in the number of correction boluses could be explained by the direction of CGM trend arrows 2 h after main evening meal. Patients in the SB treatment arm saw a mean blood sugar value of 9.1 mmol/L with a trend arrow up and therefore probably did a decision in favor to give an additional correction bolus whereas those in the DWB treatment arm saw a mean blood sugar value of 9.0 mmol/L with the trend arrow down and therefore decided to delay or skip the correction bolus. We can only speculate about the possible late postprandial blood sugar rise in SB treatment arm without additional corrections and actual late postprandial blood sugar control in the DWB group in the case of more active usage of correction boluses. Campell et al. also clearly showed that an additional 30% insulin administered 3 h after the high fat and protein meal provided additional benefit to the postprandial glucose control and made it comparable with a meal without any fat [19]. Future studies may reveal if universal utilization of the DWB with a higher initial insulin proportion as for example 70 percent initially and 30 as a extended bolus may provide better coverage for the first hour of postprandial hyperglycaemia as well as for late hyperglycaemia. Four patients benefitted from the DWB scheme for the dinner as their TIR improved more than 10% during the two weeks of DWB administration whereas in the SB arm there were two patients who improved their TIR more than 10%. Such result may be related to some unidentified random factors, but could also indicate, that personal food content preferences might play a role in this. The analysis of food content and its relation to postprandial glucose excursion in relation to different bolus types was beyond the scope of the current work as we attempted to prove the superiority of the dual-wave bolus regardless of food composition. However, it seems that the universal utilization of only one bolus type is not justified and that the choice of the bolus type should be based on the fat and protein content in the food. Nevertheless, the universal use of DWB did not increase the incidence of hypoglycemia as TBR, basal insulin suspension duration and frequency did not differ between the treatment arms. This can be postulated with a reservation as the data about prevention of hypoglycaemia by additional food was lacking, but the systematic use of the DWB for dinner seems to be a safe option in patient using insulin pump with auto suspension function.

The major limitation of our study is the small number of participants for the randomized trial and the risk for type 1 error, when making conclusions. However, to our best knowledge this is the only randomized trial comparing different types of boluses in real-life settings. Many confounders could have influenced the change in TIR and the need for correction boluses, but their impact of the confounders was diminished due to the randomized design of our study. Another significant limitation is the absence of a meal composition analysis, which should significantly influence the pattern of postprandial hyperglycemia, but this was beyond the scope of our study. Meal content analysis could have given addition insight into the benefits and disadvantages of one or the other bolus type, but our intention was to test whether the DWB is suitable for all meals typical for a Western diet [20].

## 5. Conclusions

In conclusion, the present study demonstrated that the universal DWB utilization for evening meals in which insulin is calculated solely on the food carbohydrate content did not improve TIR compared to standard bolus. The DWB did not result in lower late postprandial hyperglycemia, but DWB users used fewer correction boluses to achieve euglycemia by the morning. Future studies are needed to find the most effective bolusing type for optimal postprandial blood sugar control. 

## Figures and Tables

**Figure 1 jcm-11-01689-f001:**
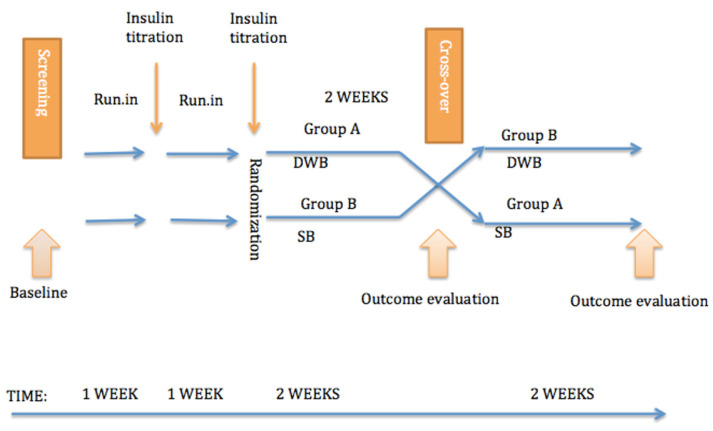
The structure of the study. DWB (dual-wave bolus); SB (Standard bolus).

**Figure 2 jcm-11-01689-f002:**
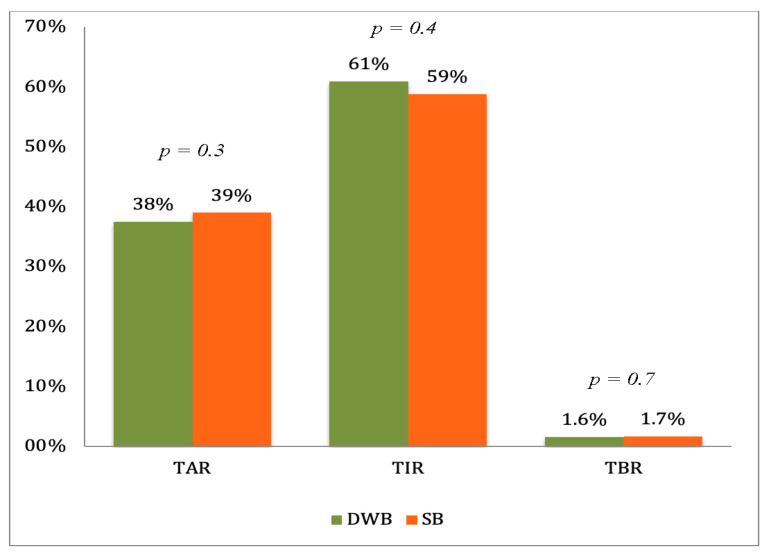
Comparison of mean TIR (Time in Range), TAR (Time above Range) and TBR (Time Below Range) between the treatment arms. DWB (dual-wave bolus); SB (Standard bolus).

**Figure 3 jcm-11-01689-f003:**
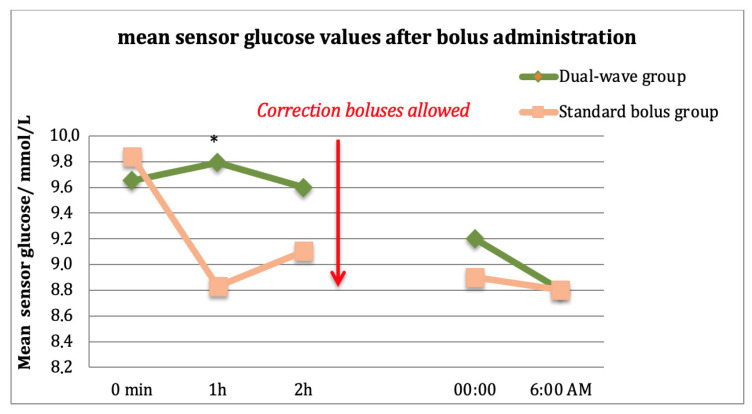
Comparison of mean sensor glucose values after main evening meal bolus administration between treatment arms. * Statistically relevant difference in sensor glucose between treatment arms, *p* = 0.01.

**Table 1 jcm-11-01689-t001:** Clinical characteristics of study subjects ^1^.

Number of Study Subjects = 24	
Gender (boys/girls)	10/14
Age (years)	10.8 ± 2.0
Weight (kg)	44.2 ± 12.6
BMI (kg/m^2^)	18.7 ± 3.4
Predicted glycated hemoglobin A1c (%)	7.6 ± 0.7
Mean sensor glucose (mmol/L)	9.5 ± 1.1
Daily insulin dose per kg (IU/kg/d)	0.8 ± 0.1
Basal insulin (%)	37.8 ± 10.7
Bolus insulin (%)	62.2 ± 10.7

^1^ Data is given as mean ± standard deviation.

## Data Availability

The trial was registered in https://clinicaltrials.gov/ct2/show/NCT04668612 under the identifier: NCT04668612.

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
