# Peer review of "Decreased Need for Correction Boluses with Universal Utilisation of Dual-Wave Boluses in Children with Type 1 Diabetes"

_jcm, 2022, doi:10.3390/jcm11061689_

Round 1
Reviewer 1 Report
In this manuscript, the authors examined ways to improve post-prandial hyperglycemia in adolescents with Type 1 diabetes during a randomized controlled crossover study comparing single boluses to the dual wave 50/50 boluses over 2hours in a real-world setting. They concluded that using DWB vs SB provided no statistically significant differences in mean TIR but a significant reduction in the needs of correction boluses. The subject is interesting and still important for consideration using the new AID systems
Main concerns
- There is no clear indication what is the primary end-point in this study. If TIR was chosen this is clearly inadequate. To examine the effects of DWB a time frame analysis of 5-6 hours should be taken using AUC.
- It is not mentioned why evening meals were chosen since no indication on meal composition especially fat contents are provided. The authors should also provide the information whether meal composition was similar during the two arm periods in order to exclude a study effect.
- The analysis of average basal suspension duration should be focused to post-prandial periods and not for the whole 14 days. Indeed the consequences of SB for fat meals are immediate hypoglycemia.
Author Response
Dear Reviewer 1
Thank you for highlighting important concerns about our research.
We have now replied to each of them point by point and revised the manuscript accordingly
Reviewer 1.
Main concerns
- There is no clear indication what is the primary end-point in this study. If TIR was chosen this is clearly inadequate. To examine the effects of DWB a time frame analysis of 5-6 hours should be taken using AUC.
We agree that TIR is not an ideal primary end-point in such a study. However, our study design where participants were encouraged to administer correction boluses two hours after the meals if the glucose value remained above 12 mmol/L, the AUC also is not a good parameter to compare treatment arms: any potential prolonged hyperglycemia that might have occurred over the next 5-6 hours after consuming a meal, was eliminated by the additional correction boluses administered. The fact that many subjects, mostly teenagers, consumed an additional meal 2-3 hours after the main evening meal, might also have a significant effect on the interpretation of AUC as an outcome measure. Therefore, in these settings TIR was chosen as the main primary end-point. Considering our study design, TIR and the description of several post-prandial glucose values with the additional information about the number of correction boluses and basal suspension episodes describes more accurately the effect of different bolus types on glycemic control in our opinion. Also, compared to the mean AUC TIR probably could better represent a clinical significance of any potential finding. Nevertheless, we calculated the mean AUC from hour one to hour 12 after the first evening meal, but did not see any statistically relevant difference. We used a time period of 12 hours because many subjects often ate repeatedly and these additional meals could heavily affect the glycemic control by influencing the blood glucose values during the whole night. We have now included the data about the mean average AUC with relevant explanations into the manuscript.
- It is not mentioned why evening meals were chosen since no indication on meal composition especially fat contents are provided. The authors should also provide the information whether meal composition was similar during the two arm periods in order to exclude a study effect.
We chose evening meals to ensure maximal adherence to study protocol because at these hours parents were able to guard compliance with the study protocol, that information is included into the method section. We have now added information in the methods and discussion sections that the subjects were encouraged to stick to their regular food composition and mealtime schedule during the two study periods.
- The analysis of average basal suspension duration should be focused to post-prandial periods and not for the whole 14 days. Indeed the consequences of SB for fat meals are immediate hypoglycemia.
The mean number of basal insulin suspensions in the 12-hour period after administering the first evening bolus and the summative mean time of suspension duration have now been reported in the Results section.
We hope that our answers to the queries and modifications made into the manuscript satisfy the reviewers and the paper can be accepted for publications.
Yours sincerely
Dr Mari Lukka

Reviewer 2 Report
In this paper authors compared efficacy of a standard bolus and double wave bolus. Article is well written and topic is interesting. I have some suggestion and some questions:
Fig1: there is a mistake (IInsulin titration). Please insert in Figure's footnote the meaning of acronyms used (SB, DWB)
IN methods authors should describe 640G characteristics, in particular PLGS function. PLGS function could mask some hypoglycemic events related to type of bolus administered. This point should be considered in discussion. Why all subjects recruited used 640G?
IN line 146 age standard deviation is not inserted.
In result section a table with participants’ characteristics should be helpful.IN line 146 authors declare that 4 subjects were excluded from analysis; this sentence should be inserted before sentence in line 144 where baseline characteristics of subjects considered were described.
I suggest to insert in results section all data described in discussion, for example in line 223 where authors comment glucose sensor values after2-6 hours.
Are there any data regarding insulin suspension in post dinner period? Are there differences in 2 groups?
In discussion authors postulated that greater number of correction boluses should be explained by arrow trend. During DWB period subjects have a mean glucose 2 h after meal greater than SB period. Did subjects were instructed to manage glucose arrows? Did they used a specific protocol to manage trend arrows?
IN line 254 authors declared that DWB didn’t increase the incidence of hypoglycemia since TBR and basal suspend time were similar. Obviously TBR was mitigate by PLGS function also in post dinner period. It would be interesting to know the data relating to insulin suspension after dinner. Are there data regarding hypo prevention treatment in post dinner period?
Obviously a limitation is the fact that meal composition is unknown, did subjects were instructed to follow a similar diet during 2 different study period? Differences from other studies that evaluated different boluses used could be explained by different meal composition?
Lines 217-220 a study performed by Lopez et al was cited but references was not correct since authors reported references 14 instead of 12
Author Response
Dear Reviewer 2
Thank you for highlighting important concerns about our research.
We have now replied to each of them point by point and revised the manuscript accordingly.
Fig1: there is a mistake (IInsulin titration). Please insert in Figure's footnote the meaning of acronyms used (SB, DWB).
This has been now done.
IN methods authors should describe 640G characteristics, in particular PLGS function. PLGS function could mask some hypoglycemic events related to type of bolus administered. This point should be considered in discussion. Why all subjects recruited used 640G?
This has been now done and discussed.
IN line 146 age standard deviation is not inserted.
This has been now done.
In result section a table with participants’ characteristics should be helpful.
A table with clinical characteristics of study subjects has been added.
IN line 146 authors declare that 4 subjects were excluded from analysis; this sentence should be inserted before sentence in line 144 where baseline characteristics of subjects considered were described.
This has been now done.
I suggest to insert in results section all data described in discussion, for example in line 223 where authors comment glucose sensor values after2-6 hours.
This has been now done.
Are there any data regarding insulin suspension in post dinner period? Are there differences in 2 groups?
These data have been now added into the Results sections. There were no significant differences regarding the time and duration of insulin suspension episodes.
In discussion authors postulated that greater number of correction boluses should be explained by arrow trend. During DWB period subjects have a mean glucose 2 h after meal greater than SB period. Did subjects were instructed to manage glucose arrows? Did they used a specific protocol to manage trend arrows?
Subjects were not instructed how to manage trend arrows. This information is now added into the Methods section.
IN line 254 authors declared that DWB didn’t increase the incidence of hypoglycemia since TBR and basal suspend time were similar. Obviously TBR was mitigate by PLGS function also in post dinner period. It would be interesting to know the data relating to insulin suspension after dinner. Are there data regarding hypo prevention treatment in post dinner period?
We do recognize that the TBR might have been mitigated by the suspend before low function activity but since there was no difference in the number of basal suspend episodes following 12 hours after the first evening meal consumption or their cumulative duration, this is unlikely. We analyzed the mean frequency of suspension episodes from 6 PM to 6 AM because many patients consumed meals at different times and many teenagers ate as late as 12 PM and made multiple boluses. Unfortunately, there were no data about hypo prevention treatment, but patients were only allowed to eat additional carbohydrates when sensor glucose was below 3,9 mmol/L during the two hours after the meal. The analysis about suspension episodes is now described in more details in the Results section.
Obviously, a limitation is the fact that meal composition is unknown, did subjects were instructed to follow a similar diet during 2 different study period? Differences from other studies that evaluated different boluses used could be explained by different meal composition?
As stated already, we agree that the lack of information about meal composition is a limitation of the study. However, the patients were encouraged to stick to their regular meal schedule and typical diet during the different treatment periods. We were able to get information about carbohydrate content in the meal from sensor glucose reports and there was no difference in carbohydrate content for dinner between the study arms. Information about fat and protein content is lacking and differences in their content may explain study results. This limitation is mentioned in the Discussion section.
Lines 217-220 a study performed by Lopez et al was cited but references was not correct since authors reported references 14 instead of 12
The citations have been now corrected.
We hope that our answers to the queries and modifications made into the manuscript satisfy the reviewers and the paper can be accepted for publications.
Yours sincerely
Dr Mari Lukka

Round 2
Reviewer 1 Report
The manuscript has improved as suggested